Organ health and development in larval kingfish are unaffected by ocean acidification and warming

Frommel Andrea Y. 1 afrommel@zoology.ubc.ca
Brauner Colin J. 1
Allan Bridie J.M. 2 3
Nicol Simon 4
http://orcid.org/0000-0002-4030-512X Parsons Darren M. 5 6
Pether Steve M.J. 7
Setiawan Alvin N. 7
Smith Neville 8
Munday Philip L. 3
1 Department of Zoology, University of British Columbia , Vancouver, BC , Canada
2 Department of Marine Science, University of Otago , Dunedin , New Zealand
3 Australian Research Council Centre of Excellence for Coral Reef Studies, James Cook University , Townsville, QLD , Australia
4 Institute for Applied Ecology, University of Canberra , Canberra, ACT , Australia
5 National Institute of Water and Atmospheric Research , Auckland , New Zealand
6 Institute of Marine Science, University of Auckland , Auckland , New Zealand
7 National Institute of Water and Atmospheric Research, Northland Marine Research Centre , Ruakaka , New Zealand
8 Oceanic Fisheries Program, Pacific Community , Noumea , New Caledonia
O’Connor Wayne
Electronic publication date: 2019 Dec 12
Publication date: 2019
Volume: 7
Electronic Location ID: e8266
Received 2019 Aug 15; Accepted 2019 Nov 21
Copyright: © 2019 Frommel et al.
Copyright year: 2019
Copyright holder: Frommel et al.
License: This is an open access article distributed under the terms of the Creative Commons Attribution License, which permits unrestricted use, distribution, reproduction and adaptation in any medium and for any purpose provided that it is properly attributed. For attribution, the original author(s), title, publication source (PeerJ) and either DOI or URL of the article must be cited.
License URL: https://creativecommons.org/licenses/by/4.0/

Keywords: CO2, pH, Temperature, Commercial fish, Seriola lalandi, Histology, Climate change, Environmental impacts, Larval development, Aquaculture

Funding: Government of New Zealand, and Principality of Monaco through the Pacific Islands Ocean Acidification Partnership (PIOAP) The Pacific Community (SPC), Australian Research Council (ARC) FT130100505 ARC Centre of Excellence for Coral Reef Studies New Zealand’s National Institute of Water and Atmospheric Research (NIWA) Funding was provided by the Government of New Zealand, and Principality of Monaco through the Pacific Islands Ocean Acidification Partnership (PIOAP), The Pacific Community (SPC), Australian Research Council (ARC), Grant/ Award Number: FT130100505, ARC Centre of Excellence for Coral Reef Studies, and the New Zealand’s National Institute of Water and Atmospheric Research (NIWA). The funders had no role in study design, data collection and analysis, decision to publish, or preparation of the manuscript.

==============================
Anthropogenic CO2 emissions are causing global ocean warming and ocean acidification. The early life stages of some marine fish are vulnerable to elevated ocean temperatures and CO2 concentrations, with lowered survival and growth rates most frequently documented. Underlying these effects, damage to different organs has been found as a response to elevated CO2 in larvae of several species of marine fish, yet the combined effects of acidification and warming on organ health are unknown. Yellowtail kingfish, Seriola lalandi, a circumglobal subtropical pelagic fish of high commercial and recreational value, were reared from fertilization under control (21 °C) and elevated (25 °C) temperature conditions fully crossed with control (500 µatm) and elevated (1,000 µatm) pCO2 conditions. Larvae were sampled at 11 days and 21 days post hatch for histological analysis of the eye, gills, gut, liver, pancreas, kidney and liver. Previous work found elevated temperature, but not elevated CO2, significantly reduced larval kingfish survival while increasing growth and developmental rate. The current histological analysis aimed to determine whether there were additional sublethal effects on organ condition and development and whether underlying organ damage could be responsible for the documented effects of temperature on survivorship. While damage to different organs was found in a number of larvae, these effects were not related to temperature and/or CO2 treatment. We conclude that kingfish larvae are generally vulnerable during organogenesis of the digestive system in their early development, but that this will not be exacerbated by near-future ocean warming and acidification.

Introduction

Increasing anthropogenic CO2 emissions to the atmosphere are leading to two major problems for marine ecosystems: rising ocean temperatures (Bindoff et al., 2007) and ocean acidification (OA) (Doney et al., 2009). OA is caused by the uptake of atmospheric CO2 at the ocean surface, where carbon dioxide is converted to bicarbonate, releasing hydrogen ions in the process. This increase in hydrogen ions has lowered the pH of the ocean surface globally by 0.1 pH units since pre-industrial times, with a further decline of 0.2–0.3 pH units predicted to occur within this century (Caldeira & Wickett, 2005). In addition to the change in seawater pH, the partial pressure of CO2 (pCO2) in the surface ocean is rising at the same rate as in the atmosphere (Doney et al., 2009). Locally, the changes in ocean chemistry caused by rising CO2 levels can be exacerbated by interacting physical and biological processes, potentially doubling the globally predicted changes in pCO2 and pH (Hofmann et al., 2011; Melzner et al., 2013; McNeil & Sasse, 2016). Concurrent to OA, sea surface temperatures have increased globally by about 1 °C since pre-industrial times and are predicted to rise 2–4 °C by the end of the century due to the enhanced green-house effect of atmospheric CO2 (Collins et al., 2013). Climate change is also increasing the frequency and duration of marine heatwaves (Oliver et al., 2018), and extreme temperatures are occuring even earlier than predicted by average projections of climate change models (Frölicher, Fischer & Gruber, 2018). OA and warming have the potential to affect the physiology and behavior of many marine organisms, leading to changes in population dynamics, community structure and ecosystem functioning (Doney et al., 2012; Gaylord et al., 2015), but the generalities and trends are still unclear, as most dynamics are inferred from small functional groups. Our ability to predict effects and make inferences for communities and ecosystems is limited by inadequate knowledge of the likely impacts of these stressors on important functional groups, such as large pelagic fishes.

Large pelagic fishes are not only important top predators for ecosystem health and food-web structure (Heithaus et al., 2008), but also contribute to 20% of global fish catch with 40 million tons per year feeding millions of people worldwide (Ye & Cochrane, 2011). Moreover, recent modeling efforts incorporating climate change predict strong effects on the distribution of large pelagic fishes, in which elevated temperature regimes may be pushing species such as tuna into predicted OA hotspots (Lehodey et al., 2013). While adult fish in general appear widely tolerant to OA due to their ability to buffer acid-base disturbances via the gills and kidneys (Brauner & Baker, 2009; Esbaugh, Heuer & Grosell, 2012), it is the early life stages that may be most vulnerable, due to the sensitivity of organogenesis and a higher surface-to-volume ratio (Ishimatsu et al., 2004; Pelster, 2008). Negative effects of elevated CO2 concentrations on growth, survival and development in early life stages of marine teleost fish have been demonstrated (Baumann, Talmage & Gobler, 2012; Coll-Lladó et al., 2018; Frommel et al., 2012; Hurst, Fernandez & Mathis, 2013), although other studies finding no effects of elevated CO2 suggest responses may be species and even population specific (Bignami, Sponaugle & Cowen, 2013; Frommel et al., 2013; Munday et al., 2011; Murray & Baumann, 2018) (see also review by Cattano et al., 2018 and ICES special edition 73, 2016).

As larvae have high overall mortality rates, they are a bottleneck in recruitment to a harvestable population (Houde, 2008), and thus represent an important life history stage to understand how environmental change may affect large pelagic species. While larval survival and growth rates offer important information for estimating climate impacts on fish population dynamics, it is equally important to understand the underlying causes for changes in condition and detect sublethal effects. Histological techniques allow for qualitative assessment of larval health and organ development and shed light on the mechanisms leading to changes in performance (Noga, 1996). Quantification of lipid content in the liver, for example, provides estimates of energy storage in the larvae and therefore growth potential (Chen et al., 2007; Sargent, Tocher & Bell, 2002). Health of the digestive system can give an indication of food conversion efficiency and condition of the larvae (Chen et al., 2006). Damage to organs, such as gills, eyes and swim bladder can impair motility, feeding and predator avoidance, leading to a decreased likelihood of survival in the wild. These are all important metrics and can be detected before changes in growth or survival take place. In addition, organ damage can be extensive even while maintaining higher growth rates than in control conditions (Frommel et al., 2012) or be sublethal (Frommel et al., 2016), leading to a significantly different evaluation of OA effects in larval fish based on growth and survival rates alone.

Yellowtail kingfish, Seriola lalandi, are mass-spawning large pelagic fishes with a circumglobal sub-tropical distribution in the southern hemisphere (Kailola et al., 1993). They are the target of both commercial and recreational fisheries and are becoming increasingly important as a species in aquaculture (Sicuro & Luzzana, 2016). As they are reliably reared in hatcheries, they are an ideal candidate to study the effects of climate changes on early life stages of large pelagic fish. In this study, we exposed yellowtail kingfish embryos and larvae to two CO2 levels (~500 µatm and ~1,000 µatm) and two temperatures (21 °C and 25 °C) in a fully crossed experimental design to investigate the effects of OA and warming on organ health and development during a critical developmental phase. The preferred water temperature for yellowtail kingfish is 21 °C (Champion et al., 2019) and is also the average summer water temperature in Northern New Zealand where this study was conducted (Shears & Bowen, 2017). The 25 °C elevated temperature treatment was chosen to match the 2–4 °C ocean warming projected under RCP 8.5 (Collins et al., 2013). Maximum water temperature can reach 23 °C in this region (Evans & Atkins, 2013); therefore, 25 °C is consistent with future summer water temperatures that may be experienced either periodically under moderate warming (2 °C) or consistently under more extreme warming (4 °C), as well as during heatwaves. Furthermore, 25 °C is sometimes used in the aquaculture of this species to elicit faster growth rates. The CO2 treatments were the current-day ambient conditions for each temperature (462 µatm and 538 µatm for 21 °C and 25 °C, respectively) and an elevated CO2 treatment of approximately 1,000 µatm CO2 to match projections under RCP 8.5 (Collins et al., 2013) (Table 1). Previous results from this experiment have demonstrated increased activity, metabolic rate and growth rate, but reduced survivorship of larval and juvenile kingfish at 25 °C relative to 21 °C (Laubenstein et al., 2018; Watson et al., 2018). By contrast, 1,000 µatm CO2 had no effect on activity, growth or survival, but did affect metabolic rate and swimming performance (Laubenstein et al., 2018; Watson et al., 2018). Thus, the overarching objective of this study was to determine whether there is a link between organ health and development and the physiological impairments seen in the early life stages of yellowtail kingfish under high CO2 and/or temperature.

Table 1 Carbonate chemistry parameters.

Mean (± SD) pCO2 for each of four treatments calculated from measured temperature (T), pH (total scale) and alkalinity using CO2SYS.

Treatment	pCO2 (µatm)	T (°C)	TA (µmol/kg SW)	pHTotal	
1	462.0 (42.8)	21.1 (0.1)	2318.8 (7.2)	8.00 (0.03)	
2	959.8 (57.3)	21.1 (0.1)	2319 (3.8)	7.72 (0.03)	
3	538.3 (15.6)	24.8 (0.4)	2319.9 (7.7)	7.94 (0.01)	
4	1010.6 (30.4)	24.9 (0.4)	2320 (6.2)	7.70 (0.01)	

Materials and Methods

Kingfish rearing

The experiment was conducted at the National Institute of Water and Atmospheric Research (NIWA) Northland Marine Research Centre in Ruakaka, New Zealand. Eggs were obtained from a broodstock of wild-caught adults (9 females and 10 males) which spawned naturally in 20 m3 circular tanks that fluctuated with the outside water temperature between 18 °C and 20 °C. Seawater for broodstock tanks was subjected to 10 μm particle filtration and then UV sterilized to 150 mws/cm2. At the time of spawning, the water was at 18.2 °C and had a pH of 7.91 (pCO2 = 589 µatm). Parentage analysis showed that 5 females and all 10 males contributed to the spawning (Munday et al., 2019). Eggs were collected on 24th January 2017 from the overflow of each of four tanks, approximately 12 h post fertilization. After disinfection with tosylchloramide (chloramine-T) at 50 ppm for 15 min and rinsing with seawater, the embryos were distributed into 24 conical incubation tanks (400 L) at an average density of 101,778 ± 9,860 (SD) eggs per tank. Seawater for egg incubation was subjected to 1 μm particle filtation (serial filtration to 10 μm, 5 μm and 1 μm) and triple (egg incubation) or double (larval rearing) UV sterilization to 150 mws/cm2. Prior to use, rearing tanks were sterilized for 30–60 min with 1% Virkon Aquatic. All handling equipment was sterilized in 300 mg/L sodium hyprochlorite solution. Footbath (1% Virkon) and handwash (1% Chlorahexidane) on entry and exit to the rearing area was enforced for all users. The incubation tanks were set in a flow-through system (3 L/min) with gentle aeration and a photoperiod of 14/10 h light/dark. After 3 h of acclimation to the incubation tanks at 18.2 °C, the temperature was slowly increased overnight to reach the two target treatment levels of 21 °C and 25 °C (see Table 1 for measured values). Any dead eggs and larvae were removed and counted daily. Larvae hatched 3 days post fertilization at 21 °C and 2 days post fertilization at 25 °C. One day post hatch (dph), larvae had 80% and 75% survival at 21 and 25 °C, respectively. Larvae 1 dph were transferred into 24 reciprocal circular tanks at an average density of 44,227 ± 2,152 (SD) larvae per tank. Grow-out tanks were 1,500 L circular tanks with slightly sloping bottoms and a black internal surface. These large grow-out tanks were kept in the same flow-through (3 L/min) and photoperiod (14/10) as the incubation tanks. We used a standard green-water protocol for larval fish rearing using Nannochloropsis (Symonds et al., 2014). Larval rearing tanks were dosed with 40 mL of Nanno 3,600 (Reed Mariculture Inc.) per day, in 2 × 20 mL doses delivered through the sump tanks that delivered water to each system. The first dose was at dawn and the second in the early afternoon to maintain green water conditions throughout daylight hours. Green-water rearing was used from 1 to 20 dph in the 21 °C treatments and from 1 to 12 dph in the 25 °C treatments. Rotifers, enriched with highly-unsaturated fatty acid and critical vitamins, were fed to the kingfish larvae at a rate of 16 mL−1 from 1 dph, transitioning to freshly hatched artemia nauplii at 11 dph. Four daily doses of rotifers were given from 3 dph. Artemia were fed in three daily doses, starting at rate of 0.33 mL−1 at 11 dph and rising to 0.90 mL−1 at 19 dph. Rotifer feeding was ceased at 12 dph at 25 °C and 19 dph at 21 °C. Dead larvae were removed by siphoning and counted daily. The surface of the water was skimmed to remove oils which may prevent larvae from gulping air for initial swim bladder inflation.

Experimental CO2/temperature system

Inflow water was pumped from the ocean, filtered through sand and fine mesh (5 µm) and UV sterilized (150 mW/cm) into large header tanks. Here, the water was enriched with oxygen (≥100%) though fine diffusers and organic material was removed with foam fractionators. Prepared seawater was then gravity-fed into eight separate sump tanks (100 L). These eight sumps were treated to create a fully crossed 2 × 2 experimental design with two replicate sumps per treatment. Each sump contained two aquarium pumps; one (HX-6540, Hailea, Guangdong, China) delivered water from the sump to the experimental rearing tanks, while the second (Maxi 103, Aqua One, Ingleburn, NSW, Australia) ensured even mixing of the water within the sump and was also the site of CO2 dosing for the elevated CO2 treatments. The pCO2 of the water was either maintained at ambient levels (4 sumps) or enriched with CO2 to achieve the target value of approximately 1,000 µatm CO2 (1,156 mg/L CO2 at 21 °C; 983 mg/L CO2 at 25 °C; pH 7.7) for the elevated CO2 treatments (4 sumps). A needle valve slowly released CO2 from a compressed CO2 cylinder into the pump inlet where it was immediately mixed by the pump impeller. The flow of CO2 to each elevated CO2 sump was regulated by a pH computer (Aqua Medic, Bissendorf, Germany) connected to a pH electrode, which dosed CO2 into the sump whenever the pH rose above the required pH setpoint for 1,000 µatm CO2. Temperature was maintained at either 21 °C or 25 °C in each sump by electronic heaters. Experimental water from each of the sump tanks was distributed to each of three replicates for a total of six replicates for each CO2 × temperature combination. Temperature and pH were measured daily in each of the rearing tanks with pH electrodes (SG8 SevenGo, Mettler Toledo, Switzerland) calibrated with TRIS-buffers (Dr. Andrew Dickson, batch number 26). Water was sampled for carbonate chemistry at 1 dph, 11 dph and 21 dph of the larvae, poisoned with mercuric chloride (at 0.05% sample volume) and later analyzed for total alkalinity and salinity by the University of Otago Research Centre for Oceanography (Dunedin, New Zealand). Salinity was 35.6 (± 0.01) during the experiment. Remaining carbonate system parameters were calculated using CO2SYS with constants from Mehrbach et al. (1973) refit by Dickson & Millero (1987) and are summarized in Table 1. For a schematic diagram of the experimental set up and more detailed methods, see Laubenstein et al. (2018) and Watson et al. (2018).

Histological processing of samples

At 11 dph a random sample of four larvae was taken from each of four tanks in each treatment (16 tanks total). At least 50% of larvae had reached flexion by this age (Watson et al., 2018). The larvae were euthanized with an overdose of 99% Isoeugenol (Aqui-S, New Zealand Ltd., Lower Hutt, New Zealand) and fixed in 4% buffered histoformalin. Four larval kingfish from each replicate were embedded pairwise in Spurr’s resin (hard), following dehydration, and cut at 1 µm with an automated ultramicrotome (Leica Ultracut 7) on a diamond knife. Every 10th section was collected onto a microscope slide, stained with Toluidine blue and mounted with an aqueous mount. Ten sections per fish were analyzed and photographed at different magnifications (10 × to 64 × oil immersion) with a microscope-mounted camera (Zeiss Axioplan Fluorescent Microscope).

The eyes, gills, liver, kidney, intestines and pancreas were investigated for possible damage from a researcher blind to the treatment from which the larvae were obtained. Organs were evaluated after the scoring system described by Bernet et al. (1999) in which the reaction pattern (regressive vs. progressive changes), the functional unit (i.e., epithelium, supporting tissue, etc.) and the type of alteration (i.e., architectural and structural alterations, atrophy and necrosis) were considered. Liver lipids were quantified using stereology. Each organ was analyzed separately and, using the formula from Bernet et al. (1999), a total lesion index was calculated for each individual. Other organs that were not targeted, such as the swim bladder and the brain, were screened for obvious pathologies.

Statistics

All statistics were performed using the open access program R® for statistical computing. Graphics were acquired with the library ggplot2. To test the effects of temperature and CO2 on the various histopathological responses of the organs, ANOVA mixed effects models were run with temperature and CO2 as fixed effects and tank as a random effect. Effect size analysis was performed using the natural logarithm of the response ratio (LnRR) and calculating a 95% confidence interval (Hedges & Olkin, 1985). To visualize the statistical distance between individuals, tanks and treatments, a multidimensional scaling plot was constructed using all organ scores for all individuals analyzed.

Ethics approval statement

Experiments were conducted under the James Cook University Animal Ethics approval number A2357.

Results

In nearly all larvae, all organs were present in the proper orientation for analysis. Figure 1 shows a representative larval section with all organs evaluated and Table 2 describes the alterations and the correlating scores for each organ. Pathologies in the eyes were mainly in the form of vacuolation of the retina (Figs. 2A–2C). The gills showed structural alterations, corrugated epithelia, aneurysms, spacing and necrosis in the lamellae (Figs. 2D and 2E). In the gut, the main alterations were in the form of vacuolization of the enterocytes and epithelial desquamation of the anterior and mid-intestine. Lipid deposits, atrophy and necrosis were less predominant lesions (Figs. 2F–2H). In the liver (a lipid storage organ) a lack of lipid deposits was the dominant alteration, with larvae in the low CO2/low temperature treatment having the largest amount of lipids, and the high CO2/high T the lowest amount of lipids (Fig. 2J). Damage to the liver was mainly detected as a granular appearance, atrophy and necrosis (Fig. 2K). The pancreas showed signs of vacuolation, change of staining characteristics, altered acinous structure and focal necrosis (Figs. 2L–2N). In the kidney, alterations were only found in the renal tubules of the head kidney, mostly in the form of cloudy swelling of epithelial cells, expansion of space inside the Bowman’s capsule and contraction of the glomerulus (Figs. 2O and 2P). In 50–90% of the larvae at 11 dph, the swim bladder was uninflated with hypertrophied epithelial cells (as shown in Fig. 1).

Figure 1 Histological section through a whole larval kingfish at 11 days post hatch.

Organs evaluated: Eye (e); gills (g); kidney (k); liver (l); pancreas (p); anterior intestine (ai); mid-intestine (mi); posterior intestine (pi); (commencement of the flexion of the notochord (n), swim bladder (sb)). Toluidine blue, magnification 10×.

Table 2 Organ damage score codes.

Description of lesions and associated damage score (following Bernet et al., 1999) for all organs evaluated. Codes can be found in histological sections, Fig. 3.

Organ	Code	Description	Score	
Eye	N	normal, no vacuoles	1	
	vac 1	1–4 vacuoles	2	
	vac 2	5–10 vacuoles	3	
	vac 3	>10 vacuoles	4	
Gills	N	normal	1	
	ab	abnormal cell structure	2	
	w	corrugated epithelial surface	3	
	sp, oe, nec	spacing, edema, necrosis	4	
Gut	N	normal	1	
	vac	vacuolization of enterocytes	2	
	epsq	epithelial desquamation	2	
	lip	lipids	2	
	atr	atrophy	3	
	nec	necrosis	4	
Pancreas	N	normal	1	
	stn	change in staining characteristics	2	
	acin	altered acinous structure	3	
	vac	vacuoles	3	
	nec	necrosis	4	
Kidneys	N	normal	1	
	cs	cloudy swelling	2	
	exp	expansion of space inside Bowman’s capsule	3	
	contr	contraction of glomerulus	3	
	nec	necrosis	4	
Liver damage	N	normal	1	
	vac	vacuolization	2	
	atr	atrophy	3	
	nec	necrosis	4	
Liver lipids	nl	no lipids	1	
	lip 1	<25% lipid vacuoles	1	
	lip 2	25–75% lipid vacuoles	2	
	lip 3	>75% lipid vacuoles	3	

Figure 2 Histological sections through various organs showing different degrees of organ damage in larval kingfish at 11 dph.

EYE: (A), (B) and (C) sections though one eye depicting vacuoles throughout the retina; GILLS: (D) normal gills; (E) gills showing corrugated epithelia, abnormal cell structure, extensive spacing and necrosis; INTESTINE: (F) normal gut with well-defined brush border of microvilli, goblet cells and food present; (G) lipid droplets overloading the midgut enterocytes, intestinal lumen filled with epithelial cells being digested; (H) vacuolization of enterocytes, epithelial desquamation, atrophy and necrosis; LIVER: (I) normal liver; (J) liver full of enlarged liver vacuoles; (K) hepatic atrophy and necrosis; PANCREAS: (L) normal pancreas; (M) vacuolation in pancreas; (N) pancreas with changed staining characteristics, altered acinous structure and focal necrosis; KIDNEY: (O) normal kidney; (P) cloudy swelling of epithelial cells of renal tubules; expansion of space inside the Bowman’s capsule and contraction of the glomerulus. See Table 2 for scoring of organ damage and associated codes. Toluidine blue, magnification 20×–40×.

There was no statistically significant effect of either CO2 or temperature, nor the combination on the type of lesions or degree of damage in any of the organs evaluated (effect size analysis of CO2 and temperature on individual organs: Fig. 3; mixed model ANOVA of temperature and CO2, including tank as a random effect, on total damage: p = 0.1909, F-statistic: 1.635 on 3 and 59 DF). An MDS plot scaling individual organ scores by treatment revealed that there was also no clustering by either temperature or CO2 treatment (Fig. 4). The only apparent effect was in the liver, which exhibited a decreasing trend in the amount and size of lipid droplets with both increasing CO2 and temperature, and a higher lesion index at elevated temperature. When corrected for fish length, the trend was even more pronounced (Fig. 5), albeit not statistically significant (lipid content of liver: mixed model ANOVA, temperature: p = 0.06, chisq = 3.5415; CO2: p = 0.15, chisq = 2.0885 and the interaction of T and CO2: p = 0.48, chisq = 0.4940; DF = 1). The hepatosomatic index was not significantly different between treatment levels (mixed model ANOVA, p > 0.05). The total lesion index for each individual larva showed higher variation between replicates within a treatment than between treatment groups (Fig. 6).

Figure 3 Effect size analysis of elevated CO2 and temperature relative to the control.

Effect size analysis of elevated CO2 relative to control CO2 at control (A) and high temperature (B); Effect size of elevated temperature relative to the control at control (C) and high CO2 concentrations (D) of lesions in different organs of larval kingfish at 11 dph using the natural logarithm of the response ratio (LnRR) and 95% confidence interval (error bars). Data is significant if the error bars do not cross the zero line.

Figure 4 Link between damage scores of all organs in individual fish for each treatment.

Multi-dimensional scaling (MDS) plot showing the statistical distance (Bray–Curtis) between individual organs scored in individual fish, tanks and treatments. Treatment 1 (light green): low CO2, low T; treatment 2 (dark green): low CO2, high T; treatment 3 (light blue): high CO2, low T; treatment 4 (dark blue): high CO2, high T. The numbers in the boxes refer to tank numbers.

Figure 5 Lipid accumulation and damage in the livers of larval kingfish.

Tukey box-whisker plot of liver lipid score with CO2 treatment at 21 °C (A) and 25 °C (B); and liver damage score with CO2 treatment at 21 °C (C) and 25 °C (D) in larval kingfish at 11 dph, normalized to standard length of individual for each treatment. Treatment 1 (light green): low CO2, low T; treatment 2 (dark green): low CO2, high T; treatment 3 (light blue): high CO2, low T; treatment 4 (dark blue): high CO2, high T.

Figure 6 Variation of total organ damage of individual larvae between tanks and treatments.

Box whisker plot showing the distribution of total damage of individual larvae for each replicate tank per treatment. Treatment 1 (light green): low CO2, low T; treatment 2 (dark green): low CO2, high T; treatment 3 (light blue): high CO2, low T; treatment 4 (dark blue): high CO2, high T.

Discussion

There were no significant effects detected of either elevated CO2 or warming, nor the combination, on the organ health and development in 11 day old kingfish larvae. While lesions were found in the eye, gills, intestine, kidney, liver and pancreas of some larvae, we did not detect any relationship with the experimental treatment. Mortality rates were consistent with rates for this and other kingfish hatcheries and the broodstock, eggs and larvae used in this study seemed a normal, healthy batch.

Elevated CO2 concentrations have significant effects in some or all of these tissues in larvae of Atlantic cod (Frommel et al., 2012), Atlantic herring (Frommel et al., 2014) and yellowfin tuna (Frommel et al., 2016), at comparable developmental stages. However, the lowest CO2 concentrations tested in these previous experiments was 1,800 µatm for Atlantic cod and herring, and 2,000 µatm for yellowfin tuna. It may be that effects at 1,000 µatm are subtle and unlikely to impact normal organ development for most fishes, while 1,800 µatm may pose a tipping point for larval fish physiology. This is an area clearly worthy of further investigation. On the other hand, 1,100 µatm CO2 was found to double daily mortality rates in Atlantic cod (Stiasny et al., 2016), therefore, yellowtail kingfish may be more tolerant to elevated CO2 than these other species. Whether temperature mediates, exacerbates or has no effect on organ health and development at higher CO2 levels remains unknown as previous experiments were not cross-factored with elevated temperature.

A recent meta-analysis by Cattano et al. (2018) found the eco-physiological responses of fish to OA to be influenced by latitude. In their analysis, overall mortality was increased and growth reduced only in subtropical pelagic species, while tropical, temperate and polar species showed no overall directionality of response. Both Atlantic cod and herring are temperate species (one pelagic, one benthic) that spawn in waters between 5 °C and 10 °C, while yellowfin tuna are a tropical pelagic species that spawn around 25 °C. Since all three species were found to have mild to severe damage to internal organs following chronic elevated CO2 exposure, we can conclude that OA has the potential to impact organ health over a wide range of latitudes and temperature regimes, at least above 1,800 µatm CO2. The subtropical pelagic yellowtail kingfish, on the other hand, seem to be robust to elevated CO2 conditions, even at temperatures above their thermal preference. Adult kingfish prefer water temperatures between 18 °C and 24 °C, with peak abundances close to 21 °C (Champion et al., 2019). Kingfish populations exhibit strong seasonal changes in their distribution in response to water temperature, with abundances declining sharply at temperatures above 22.5 °C (Brodie et al., 2015). Optimal rearing temperature for kingfish larvae in hatcheries is around 21–22 °C (Fielder & Heasman, 2011), suggesting that this is also the adaptive temperature for larval performance and recruitment to the adult population. Water temperatures above the apparent thermal optima did not appear to alter the sensitivity of larval yellowtail kingfish to elevated CO2. The absence of CO2 effects on organ health and development observed here is consistent with the absence of elevated CO2 effects on growth and development of the whole animal at either 21 °C or 25 °C (Watson et al., 2018).

Previous results have demonstrated that temperature had a greater effect on survival and growth of kingfish larvae at 11 dph than elevated CO2. While mortality was negatively correlated with temperature, and growth was positively correlated with temperature, 1,000 µatm CO2 did not affect survival, growth or external morphology relative to ambient controls (Watson et al., 2018). Furthermore, there were no effects of CO2 on behavioral traits such as activity and boldness. However, both elevated CO2 and temperature did lead to an increase in resting oxygen uptake relative to the control (Laubenstein et al., 2018). This suggests that larvae were expending more energy at rest in the elevated CO2 and temperature treatments, than the control animals. This is consistent with our liver histology results, where we found a (statistically non-significant) trend towards decreased lipid contents in the elevated CO2 and temperature treatments, indicating that the larvae did not have the same energy storage capacity as the control animals. In the wild, even slightly lower energy stores and higher metabolic demands can have significant effects on larval survival (Llopiz et al., 2014). Indeed, both higher temperature and elevated CO2 in combination with food limitation has been found to affect survival, growth and developmental rate of larval fishes (Gobler et al., 2018; McLeod et al., 2013).

Lesions throughout the tissues we examined were found in larvae from all treatments; however, unlike the other studies on OA and organ damage (Frommel et al., 2012, 2014, 2016), the lesions were unrelated between tissues. In the cod, herring and tuna, the likelihood of damage to one organ increased with damage found in another. This was not the case in the kingfish, where damage between organs and larvae were distributed randomly. The large variation in organ health within treatments was quite striking in this study and was larger than the variation between treatments. Analysis of the tank effect showed that only about 7% of the variation could be explained by tank alone, the same percentage variation that was explained by CO2 and temperature. This indicates that the variation within the tanks was high and individual larval variation must explain the remaining 93%. This could be due to the large number of adults contributing to the spawning event, and the resulting variation in offspring fitness carried over from the adults. Maternal provisioning has been shown to influence offspring size, metabolic rate and condition and thus drive phenotypic variation in intrinsic factors, leading to large differences in survival, even under optimal rearing conditions (Johnson et al., 2014).

The retinal lesions and swim bladder non-inflation may be indicative of viral infections. Betanodaviruses, for example, cause retinopathy and are linked to anorexia and lethargy. However, these viruses typically also cause brain vacuolization (Munday & Nakai, 1997), which was not found in this study. While brains were not the main focus of this study, they were screened for obvious pathologies. Also, much care was taken to sterilize the rearing water and eggs and larvae are regularly screened for a range of pathogens, including nodavirus, birnavirus, novirhabdovirus, rhabdovirus, Oncorhynchus masou virus, infectious and epizootic hematopoietic necrosis viruses and infectious salmon anemia from orthomyxovirus (Biosecurity New Zealand Diagnostics Laboratory, Wellington, New Zealand). All samples, including the eggs and larvae from the broodstock used in this experiment, have returned negative results for all screened viruses, indicating that eggs and larvae from these fish were free from viral infection. The gill lesions are not consistent with viral infection, where hyperplasia and lamellar fusion are typical pathologies, whereas necrosis in the gills is more consistent with a toxicological response. No toxins were detected in our experiment. If the pathologies in our study were linked to a viral infection, neither temperature nor CO2 exacerbated the severity of infection, which would be an interesting finding in itself and worthy of further investigation.

Non-inflation of the swim bladder is typical in kingfish hatcheries and while individuals without inflated swim bladders can survive, this condition is related to high mortalities in the culture of this species (S. Pether, 2018, pers. obs.; Woolley & Qin, 2010). Much effort is being undertaken to understand why hatcheries see such high rates of swim bladder non-inflation in larval kingfish, yet the factors affecting inflation rates remain poorly understood and so-far no single cause has been found (Woolley & Qin, 2010). A possible explanation for swim bladder non-inflation in our experiment may be the link between the swim bladder and the digestive system. Yellowtail kingfish are physostomes which inflate their swim bladder by gulping air from the surface (Woolley & Qin, 2010). This air then moves from the gut through the pneumatic duct into the primordial swim bladder for initial inflation. The window for successful inflation is very small (3–5 dph) after which the pneumatic duct degenerates and closes. Lesions to the digestive tract could prevent successful swim bladder inflation during this sensitive developmental phase.

The lesions to the digestive tract found in this study are consistent with larval starvation (Chen et al., 2007), notably those related to digestion in the intestine, the liver and the pancreas. It has been shown that kingfish larvae are especially vulnerable to starvation in the early developmental phase (<15 dph) (Chen et al., 2007) and that this can occur even at abundant food supply in culture due to poor vision and mouth gape limitation (Planas & Cunha, 1999) and difficulties in digestion (Chen et al., 2006). Organ health is likely to improve in older larvae that have a fully developed gastric gland and pyloric ceca that enables acid digestion of live food (Chen et al., 2006; Partridge, 2013). Due to technical constraints, larvae at 25 dph were not sectioned and analyzed, however, in juvenile kingfish no effects were found on survival, food intake or blood parameters at water pH levels as low as 7.16 and temperatures between 21 °C and 26.5 °C (Abbink et al., 2012) indicating that low environmental pH does not affect feeding or digestive physiology in juveniles.

Conclusions

Our results suggest that the early life stage around 11 days post hatch is particularly vulnerable for kingfish larvae and leads to considerable variation in organ health among individuals consistent with high mortality rates. However, organ health of larval kingfish does not appear to be negatively affected by near-future temperatures and CO2 levels. Therefore, we conclude that kingfish organ development is robust to the effects of ocean warming and acidification predicted to occur within this century.

Supplemental Information

Supplemental Information 1 Raw data for histological assessment of kingfish larval health.

Descriptive assessment of alterations in all major organs analyzed in histological sections of all kingfish larvae at 11 dph; Codes described for scoring alterations in each organ; Scores given to each organ in each fish based on the code previously described.

Click here for additional data file.

The authors would like to thank staff at the Northland Marine Research Centre for assistance with running the experiment and collecting samples, and technicians at the Bioimaging Facility at UBC for providing technical assistance in histological processing.

Additional Information and Declarations

Competing Interests

Author Contributions

Animal Ethics

Data Availability

The authors declare that they have no competing interests. Darren Parsons is employed by the National Institute of Water and Atmospheric Research (NIWA). NIWA is a wholly government owned company, that conducts environmental research, often for government ministries, but also for private companies.

Andrea Y. Frommel analyzed the data, contributed reagents/materials/analysis tools, prepared figures and/or tables, authored or reviewed drafts of the paper, approved the final draft.

Colin J. Brauner analyzed the data, contributed reagents/materials/analysis tools, authored or reviewed drafts of the paper, approved the final draft.

Bridie J.M. Allan performed the experiments, authored or reviewed drafts of the paper, approved the final draft.

Simon Nicol conceived and designed the experiments, authored or reviewed drafts of the paper, approved the final draft.

Darren M. Parsons conceived and designed the experiments, performed the experiments, authored or reviewed drafts of the paper, approved the final draft.

Steve M.J. Pether conceived and designed the experiments, performed the experiments, authored or reviewed drafts of the paper, approved the final draft.

Alvin N. Setiawan performed the experiments, authored or reviewed drafts of the paper, approved the final draft.

Neville Smith conceived and designed the experiments, authored or reviewed drafts of the paper, approved the final draft.

Philip L. Munday conceived and designed the experiments, performed the experiments, analyzed the data, contributed reagents/materials/analysis tools, authored or reviewed drafts of the paper, approved the final draft.

The following information was supplied relating to ethical approvals (i.e., approving body and any reference numbers):

This study was approved by the James Cook University Animal Ethics Committee (approval number A2357).

The following information was supplied regarding data availability:

The raw data is available in the Supplemental File.

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
