# Peer review of "Organ health and development in larval kingfish are unaffected by ocean acidification and warming"

_PeerJ, doi:10.7717/peerj.8266_

## Round 0.1 · original submission · Minor Revisions

Both reviewers have commented on the quality of the paper but have suggested a number of useful amendments.

Minor corrections:

"Fielder" and Heasman (correct in text and refs)
L399 species in italics
Check abbreviations are used consistently eg L156 mL

Reviewer 1 ·

Basic reporting

The English used in the MS is excellent. One thing that could be improved is removal of redundant statements such as "Previous studies have found significant effects of elevated CO2 concentrations in some or all of these tissues" which could be edited to "Elevated CO2 concentrations have significant effects on some or all of these tissues" and use of "as well as" where "and" would suffice.

The background, references and context are very good, and more than adequate. The structure is excellent and the presentation is appropriate. Data are available and the MS is self-contained.

Experimental design

These are all appropriate for the contents and findings.

Validity of the findings

The findings in general are excellent.

The most important factor in larval development of kingfish that manifests in hatchery culture is development of the swim bladder and adequate vascularisation of the swim bladder. Swim bladder pathologies are described in the background, but are not examined in the study. If this can be analysed it should be, but the choice of the organs and why they were chosen should also be described.

Vacuolation of the retina is consistent with viral nervous necrosis caused by betanodaviruses. These are common in hatcheries and are recorded from carangids such as Pseudocaranx dentex (silver trevally/striped jack) - the striped jack nervous necrosis virus causes losses in hatchery culture of striped jack in Japan. These viruses typically also cause brain vacuolation - were the brains examined? Were the fish screened for nodaviruses? If any materials remain they should examined for these viruses.

Mortality and the "normality" of the batch is not described. This needs to be established to ensure that the findings are valid.

Additional comments

Line 201 - Aqui-S is not 50% isoeugenol (it is 99+% isoeugenol). It is better to cite it as a proprietary product and name the manufacturer.

Reviewer 2 ·

Basic reporting

This is a very well written manuscript. The experiment appears solid and the histology expertly done. I did not find many issues and I think this will be a nice addition to the ocean acidification literature. Publication of negative results doesn't happen enough. Negatives results are generally more difficult to publish, and much research ends up unpublished, so it was nice to read this manuscript. It is also positive to see a fish larvae not seriously affected by CO2 and warming.

The manuscript is well referenced.

The figures are very good with boxplots, lnRR, MDS, and pretty histology. I wonder if figure 2 could be nicer in colour? What is the reason it is presented in black and white?

Experimental design

The experimental design is is good but it may have an issue. The warm groups were heated 7°C over 24 hours. That means the warm group got an acute heat shock that the two cold groups didn't get. Normally, chronic thermal exposures are applied gradually to the treatment groups so that the factor is chronic and not a confounded mix of acute and chronic. If there are references showing that the embryos are thermally tolerant, then that should be presented. Otherwise this potential issue should be acknowledged.

Validity of the findings

The negative results are compared with the relevant literature and the interpretations are sound.

Additional comments

Minor comments for the authors:

Line 100. Higher than what?

Line 122. Why did the two control treatments differ in pCO2. That is not predicted by the ipcc models.

Line 148. An acute seven degree increase sounds potentially damaging. Was this rate of warming validated? This potentially harmful heat shock was applied to only one of the temperatures and could thus have introduced a confound.

Line 210. Great to see blinded scoring used here.

Line 205. How many sections were scored per fish?

Line 223. LnRR is a great way of visualizing effect sizes and isn’t used nearly enough, so I’m pleased to see that used here.

Line 267. All four treatments had temperature and co2, so their "combination" doesn’t make sense here. Elevated co2 and warming, and the combination would make sense.

284. I don’t understand “eco physiology of fish to OA”. Rephrase.

293. Preference and physiologically optimal temperatures can differ so it’s not certain that 21 is the more optimal temperature. Especially as the high temperature is used in aquaculture of the species.

Line 313. The authors discuss non-significant trends and while some may advice against that, I think it’s correct in this case. The binary thinking of significance testing should be a thing of the past.

Line 229. Why was this under a JCU ethics permit? Shouldn’t the host institute and host country review the ethics of their experiments?

---

## Round 0.2 · Minor Revisions

Please consider the suggested minor suggestions.

Reviewer 1 ·

Basic reporting

The ms is now mostly suitable for publication.

Experimental design

The experiment is well designed.

Validity of the findings

The findings are valid, but the authors need to describe the “water sterilisation” if they are relying on it for assurance the fish were virus free. Also, nodaviruses are often pseudo vertically transmitted, so the status or surveillance of the parents should be described, or it should be stated that their status is unknown.

Additional comments

The ms is much improved but could benefit from a final edit for flow and sense.

---

## Round 0.3 · accepted · Accept

Thank you for your efforts to accommodate the reviewer suggestions, this work will be a valuable contribution to the literature.